Methods

# Silica-based solid-phase extraction of cross-linked nucleic acid–bound proteins

Claudio Asencio[1,2] , Aindrila Chatterjee[1] , Matthias W Hentze[1]

**Proteins interact with nucleic acids to regulate cellular functions. The study of these regulatory interactions is often hampered by the limited efficiency of current protocols to isolate the relevant nucleic acid–protein complexes. In this report, we describe a rapid and simple procedure to highly enrich cross-linked nucleic acid–bound proteins, referred to as "2C" for "complex capture." This method is based on the observation that silica matrix–based columns used for nucleic acid purification also effectively retain UV cross-linked nucleic acid–protein complexes. As a proof of principle, 2C was used to isolate RNA-bound proteins from yeast and mammalian Huh7 cells. The 2C method makes RNA labelling redundant, and specific RNA–protein interactions can be observed and validated by Western blotting. RNA–protein complexes isolated by 2C can subsequently be immunoprecipitated, showing that 2C is in principle compatible with sensitive downstream applications. We suggest that 2C can dramatically simplify the study of nucleic acid–protein interactions and benefit researchers in the fields of DNA and RNA biology.**

## Introduction

From storage and transmission of genetic information, in the form of DNA and RNA, to the regulation of gene expression by silencing mechanisms and effector functions like ribozymes, nucleic acids play central roles in cellular life. For decades, intense research efforts were directed to the development of simple methods to isolate pure nucleic acids from diverse biological origins. Key discoveries for the implementation of simple nucleic acid isolation protocols were the strong protein denaturing properties of chaotropic salts (Gordon, 1972; Chirgwin et al, 1979), the differential partitioning of nucleic acids and proteins in aqueous and organic phases as a single step method for nucleic acid purification (Chomczynski & Sacchi, 1987), and the discovery of the strong and selective binding of nucleic acids to silica-based matrices (Boom et al, 1990; Koo et al, 1998). These innovations enormously simplified the isolation of nucleic acids and led to the development of protocols and commercial kits that are being widely used.

Biologically, the cellular roles of nucleic acids can only be understood in the context of proteins that assist and accompany them throughout their existence (Muller-McNicoll & Neugebauer, 2013; Rafiee et al, 2016). Different protocols have been developed to isolate DNA– and RNA–protein complexes, to study their composition and biological functions. Although purification under native conditions better preserves biological complexes in their physiological states, it usually suffers from profound contamination by other cellular components (Gagliardi & Matarazzo, 2016). Alternatively, cross-linking can be employed to stabilize nucleic acid–protein interactions, allowing more stringent conditions to be used during purification to reduce the levels of contaminants. In this way, several methods to study DNA–protein interactions involve the usage of cross-linkers to stabilize DNA–portein complexes, followed by chromatin fragmentation and immunoprecipitation of the protein of interest, as employed in the different chromatin immunoprecipitation protocols that are currently available (Orlando, 2000; Park, 2009; Rafiee et al, 2016).

The study of RNA–protein complexes, or RNPs, faces additional technical challenges derived from both the lability of RNA and the more dynamic nature of RNA-based processes within cells (Muller-McNicoll & Neugebauer, 2013; Dassi, 2017). One of the preferred cross-linking methods to stabilize RNA–protein interactions in vivo is UV light (Wagenmakers et al, 1980), which promotes the formation of covalent bonds between RNAs and directly bound proteins at "zero distance" (Hockensmith et al, 1986; Brimacombe et al, 1988). Several techniques to study RNPs based on UV cross-linking have paved the way to expanding our knowledge on RNA metabolism (Hentze et al, 2018). The development of interactome capture and adaptations of this protocol allowed the generation of comprehensive lists of RNA-binding proteins (RBPs) in different organisms and facilitated comparative studies between different conditions (Baltz et al, 2012; Castello et al., 2012, 2013; Beckmann et al, 2015; Hentze et al, 2018). The identification of the peptide domains directly involved in RNA binding benefited from the development of RBDmap (Castello et al., 2016, 2017). Finally, target RNA molecules bound by a specific RBP can now be determined by a suite of cross-link and immunoprecipitation-based protocols (Van Nostrand et al, 2016; Lee & Ule, 2018).

[1]European Molecular Biology Laboratory, Heidelberg, Germany  [2]Centro Andaluz de Biología del Desarrollo, Universidad Pablo Olavide, Sevilla, Spain

Correspondence: asencio@embl.de; hentze@embl.de

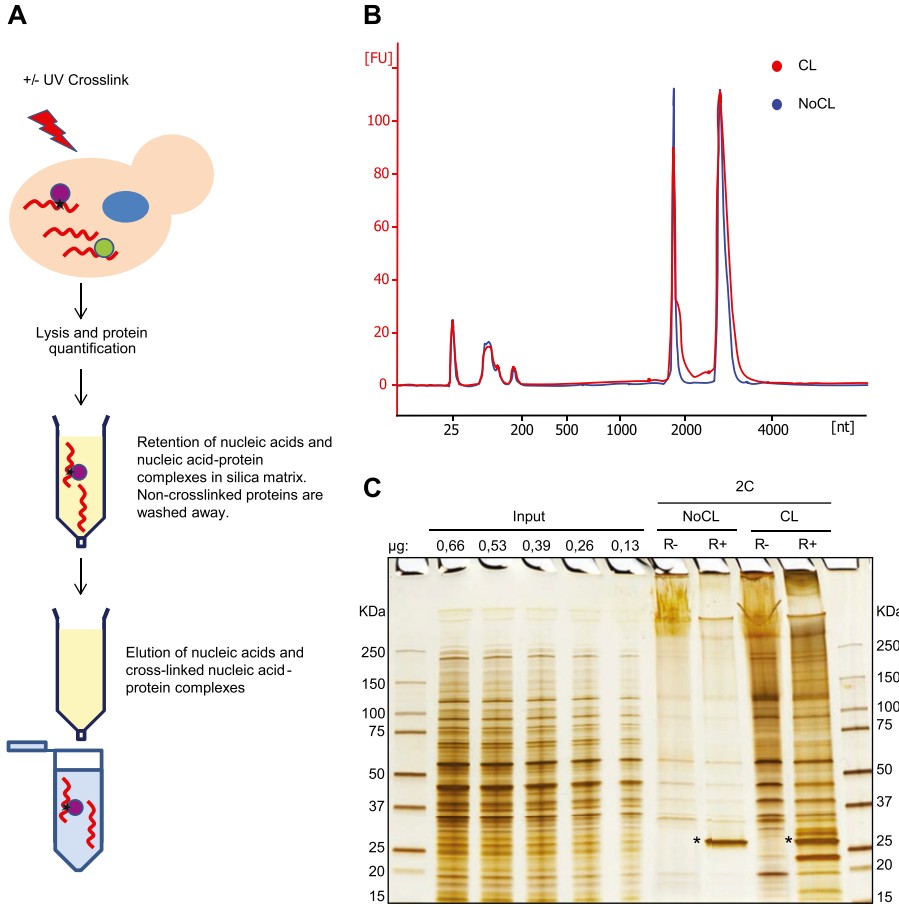

**A**

+/- UV Crosslink

Lysis and protein quantification

Retention of nucleic acids and nucleic acid–protein complexes in silica matrix. Non-crosslinked proteins are washed away.

Elution of nucleic acids and cross-linked nucleic acid-protein complexes

**B**

**C**

**Figure 1. Isolation of RBPs from *S. cerevisiae* by 2C.**
**(A)** Schematic representation of the 2C method. **(B)** Analysis of RNA integrity of cross-linked (red) and non–cross-linked (blue) yeast samples after irradiation of the cells with 3 J/cm² of UV light at 254 nm and 2C extraction. **(C)** Visualization of yeast RBPs. 2C eluate samples equivalent to 12.5 μg of RNA were treated or not with RNase I, boiled for 5 min in loading buffer, and separated through a 4%–15% gradient SDS–PAGE subjected to silver staining. We noticed that the background contamination seen in the "NoCL" samples can be virtually eliminated by preincubation of the lysates at 70°C for 5 min. R−, non–RNase-treated samples; R+, RNase I–treated samples; *RNase I protein.

Here, we report that a silica-based solid-phase extraction, developed for the purification of nucleic acids, can be used to capture individual proteins cross-linked to nucleic acids or complete nucleic acid–bound proteomes. We refer to this simple and robust approach as "complex capture," or 2C. The 2C method dramatically simplifies the isolation of nucleic acid–protein complexes, does not require the use of radioactivity for the detection of specific nucleic acid–binding proteins, and simplifies downstream applications.

## Results

### Capture of cross-linked RNPs using silica columns

Because silica columns are used to retain and purify DNA and RNA based on charge, we wondered whether they might also retain nucleic acid–binding proteins when covalently cross-linked to nucleic acids. The presence of strong denaturing agents would prevent the retention of non–cross-linked proteins and only cross-linked proteins or peptides would be retained indirectly via their bound nucleic acid. These nucleic acid–binding proteins could later be co-eluted with pure DNA, RNA, or both (Fig 1A).

We decided to test this idea for RBPs and used yeast as well as human cells for proof-of-principle experiments. *Saccharomyces*

*cerevisiae* cells were cross-linked by irradiation with 3 J/cm² UV light at a wavelength of 254 nm. Equivalent samples of non-irradiated cells were used in parallel as negative controls. Cell lysates were treated with a guanidinium thiocyanate–containing buffer, which is commonly used for RNA purification. Under these conditions, proteins are fully denatured, and the binding of RNA to the silica matrix relative to DNA is favored by the addition of ethanol (Avison, 2008). The samples were applied to silica columns, followed by intensive washing and elution into water.

After elution, we analyzed the integrity of the RNA and found that neither the cross-linking nor the purification procedure compromised RNA integrity (Fig 1B). To explore the presence of cross-linked RBPs in the eluates, samples were either treated with or without RNase I, separated by SDS–PAGE, and visualized by silver staining (Fig 1C). In contrast to the non–cross-linked controls, cross-linked samples displayed a range of bands across the complete spectrum of molecular masses, suggesting strong and selective enrichment of RBPs. This general pattern persists after extensive treatment with RNase I, demonstrating that the diversity of the bands observed in the cross-linked sample results from proteins. Moreover, the pattern of cross-linked proteins is quite distinct from the input material, implying that the retained proteins are mostly specific RBPs rather than abundant cellular contaminants (Fig 1C). These results strongly support the notion that cross-linked RBPs can be retained on a silica matrix, purified, and co-eluted with nucleic acids.

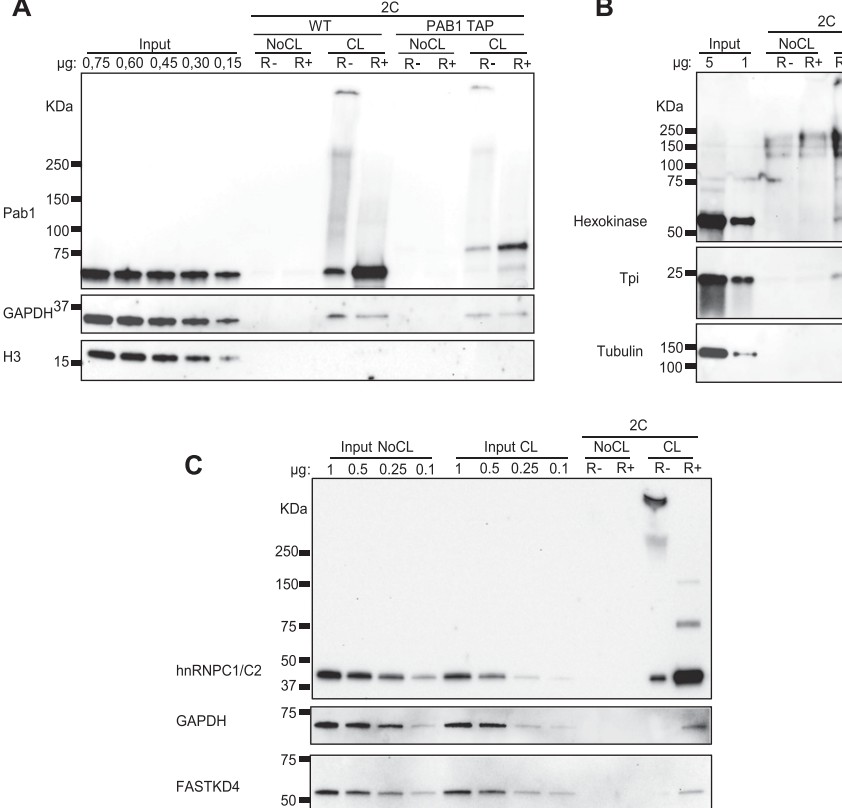

**Figure 2.  Validation of specific RNA–protein interactions from yeast and mammalian cells isolated by 2C.**
**(A)** Evaluation of 2C performance examining known yeast RBPs by Western blotting. 2C eluates equivalent to 12.5 µg of RNA were treated or not with RNase I, boiled in loading buffer, and separated through SDS–PAGE. Specific RBPs were visualized with antibodies against Pab1 and GAPDH. The DNA-binding histone H3 was probed as a negative control. **(B)** Validation of additional non-canonical yeast RBPs by Western blot after 2C. The samples were treated as in (A), and hexokinase, triose phosphate isomerase, and tubulin, as a negative control, were analyzed with specific primary antibodies. Note that *S. cerevisiae* hexokinase B is known to aggregate under denaturing conditions in vitro into amyloid-like fibrils (Ramshini et al, 2011). The denaturing conditions during 2C capture, thus, may have promoted the formation of hexokinase aggregates resistant to SDS–PAGE separation. **(C)** Analysis of mammalian 2C eluates by Western blot. The samples were treated as above (A), and the proteins hnRNPC1/C2, GAPDH, FASTKD4, and histone H3 were detected with specific primary antibodies. R–, non–RNase-treated samples; R+, RNase I–treated samples.

## Analysis of specific RBPs validates 2C-based enrichment

To validate the enrichment of RBPs in the eluate of cross-linked samples, Western blot experiments were conducted to examine specific proteins. Similar to the silver gel shown in Fig 1C, the samples were conditionally treated with RNase I before gel loading. We first examined an abundant, high-affinity RBP, the poly-A binding protein (Pab1). In addition to the WT yeast strain, a PAB1 tandem affinity purification (TAP)–tagged strain was used in parallel. As expected, we detected an intense signal in the UV cross-linked samples in contrast to the untreated negative controls (Fig 2A). TAP–tagged Pab1 proteins displaying slower migration were also observed in the corresponding cross-linked samples, demonstrating that both forms of Pab1 were retained by the silica matrix in a cross-linking–dependent way.

The samples that were not treated by RNase I showed a continuous upward smear from the expected Pab1 band, which collapsed to the expected size of Pab1 following treatment with RNase I, strongly suggesting that the smear reflects RNA-bound Pab1 (Fig 2A). T4 polynucleotide kinase assays are typically used to assess RNPs by radioactive labelling. Our results show that the 2C method can be used without radioactive labelling to detect and validate RBPs in a simple step.

Next, we tested the 2C method to probe for a weaker, non-canonical RNA binder. As yeast GAPDH (Tdh3) has been identified as an RBP in multiple interactome capture studies, we selected it as

our next candidate. As shown in Fig 2A, GAPDH was selectively detected in the cross-linked samples.

Of note, comparison of input signals with the 2C eluates allows assessment of the RNA binding capacity of different RBPs, although the amount of RNA-bound protein, compared with its overall abundance, is also influenced by its ability to be cross-linked to RNA. As expected, Pab1 behaves as a potent RNA binder and/or displays the features of a protein with high RNA cross-linking efficiency. Conversely, a much smaller fraction of GAPDH appears to be bound to RNA for either or both of the reasons discussed above. Analysis of the DNA-binding histone H3, used here as a negative control, demonstrates the limits of possible DNA contamination in our RNA purifications (Fig 2A).

Similarly, the yeast glycolytic enzymes hexokinase (Hxk) and triose phosphate isomerase (Tpi), which were previously identified as RBPs (Castello et al, 2012; Beckmann et al, 2015), were also specifically enriched by 2C (Fig 2B). Tubulin was used in this experiment as an additional specificity control (Fig 2B).

We next evaluated the performance of 2C in mammalian cells. To this end, we used human liver carcinoma-derived Huh7 cells. Similar to the yeast methodology, Huh7 cells were cross-linked in vivo and the cell lysates were subjected to the 2C protocol. The proteins hnRNPC1/C2 were analyzed as examples of strong RBPs (Fig 2C). Similar to Pab1 in yeast, a slowly migrating band of hnRNPC1/C2 collapsed to the expected size of the protein following treatment with RNase I (Fig 2C). Mammalian GAPDH and the

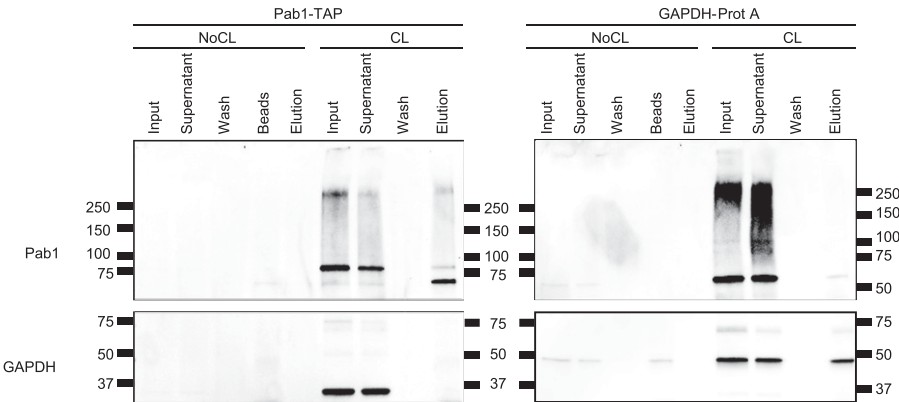

**Figure 3. Affinity purification of Pab1–RNA and GAPDH–RNA complexes after 2C.**
RNPs of PAB1-TAP and GAPDH-ProtA strains were isolated by the 2C method. Eluates from 2C were subsequently affinity purified via the tags. Fractions from input, supernatant, washes, and elution were collected, subjected to Western blot analysis, and probed with antibodies against Pab1 and GAPDH.

non-canonical mitochondrial RBP FASTKD4 (Wolf & Mootha, 2014; Boehm et al, 2017) were additionally tested along with histone H3 as a negative control. Thus, the data shown in Fig 2C provide further experimental support that 2C can be used to discover and validate RNA–protein interactions.

### Affinity purification of RNPs after 2C

2C involves protein denaturation for stringent removal of non–cross-linked polypeptides. As a consequence, proteins in the eluate may lack their native conformation hindering the subsequent immunoprecipitation of specific native or tagged RBPs and their bound RNAs. Therefore, we tested tagged RBPs in affinity purification experiments after 2C capture.

Tdh3, corresponding to isoform 3 of yeast GAPDH, and Pab1 were tagged with the protein A or the TAP tag, respectively. 2C eluates were used as inputs for affinity purification experiments, and aliquots from input, supernatant, wash, and eluate fractions were tested by Western blotting. As expected, little or no protein was detected in the lanes corresponding to non–cross-linked samples (Fig 3). Both Pab1 and GAPDH are detected in the input fractions of the cross-linked samples, showing that tagging did not compromise the RNA binding capacity of these proteins. Importantly, the tagged proteins are detected in the eluates, reflecting their affinity purification after the 2C protocol (Fig 3). These results show that 2C eluates in principle are suitable for downstream affinity purification protocols, expanding the possible applications of 2C.

## Discussion

The main finding of this study is that silica matrices are not only capable of retaining nucleic acids but also polypeptides that are cross-linked to them. Here, we used RNA and RBPs to demonstrate the validity of the 2C technique. This seemingly trivial finding has profound potential to advance experimentation in RNA biology, especially of RNA–protein interactions. Because silica binding of DNA follows the same fundamental principle, we expect that 2C is equally applicable to DNA-binding proteins.

One of the popular methods to identify and study RNA–protein interactions is UV cross-linking, which, however, suffers from the

low efficiency of covalent bond formation between nucleic acids and proteins (Darnell, 2010). As a consequence, only a small fraction of existent complexes are stabilized by a UV dose that does not compromise nucleic acid integrity. Several approaches have been devised to overcome this limitation. RNA interactome capture immobilizes polyadenylated RNA on a solid support of oligo (dT) beads. In this way, mRBPs can be specifically and highly enriched, but the procedure fails to retain RBPs bound to non-polyadenylated RNAs. To overcome this limitation, the chemistry-assisted RNA interactome capture protocol is based on the incubation of cells with two nucleotide analogs that will be incorporated into nascent RNA molecules (Huang et al, 2018). 4-thiouracil more effectively cross-links to proteins after irradiation at 365 nm wavelength, whereas 5-ethynyluridine can be biotinylated by "click" chemistry, allowing its capture on streptavidin-coated beads. In this way, all forms of RNAs can be immobilized on a solid support and cross-linked proteins be purified under stringent conditions. The main disadvantages of this method derive from the toxic effects of the nucleotide analogs on living cells and the labor-intensive manipulation of the samples needed for the purification of RBPs.

In contrast, the 2C method is based on a simple, well-established principle: the inherent property of silica matrices to strongly and specifically retain nucleic acids (Boom et al, 1990; Koo et al, 1998). This interaction is sufficiently strong to retain even larger RNA–protein assemblies, as evident by the detection of high molecular weight proteins on silver-stained gels (Fig 1C) and Western blots (Figs 2 and 3).

An additional advantage of 2C is the simplicity of detection and validation of specific RBPs. The conventional protocol to validate RBPs involves immunoprecipitation, stringent washing, and subsequent radiolabeling of cross-linked RNA with T4 polynucleotide kinase assay. The RBP is then eluted, resolved by SDS–PAGE, and detected by autoradiography. Sensitivity of the signal to RNase treatment supports the conclusion that the immunoprecipitated protein is an RBP (Baltz et al, 2012; Kwon et al, 2013; Beckmann et al, 2015). By contrast, the 2C method does not require immunoprecipitation or radiolabeling for the validation of an RBP. A simple Western blot can be used to detect and validate several RBPs in parallel, as shown in Fig 2. Thus, we have experienced that 2C simplifies the use of both materials and methods required for the study of RBPs.

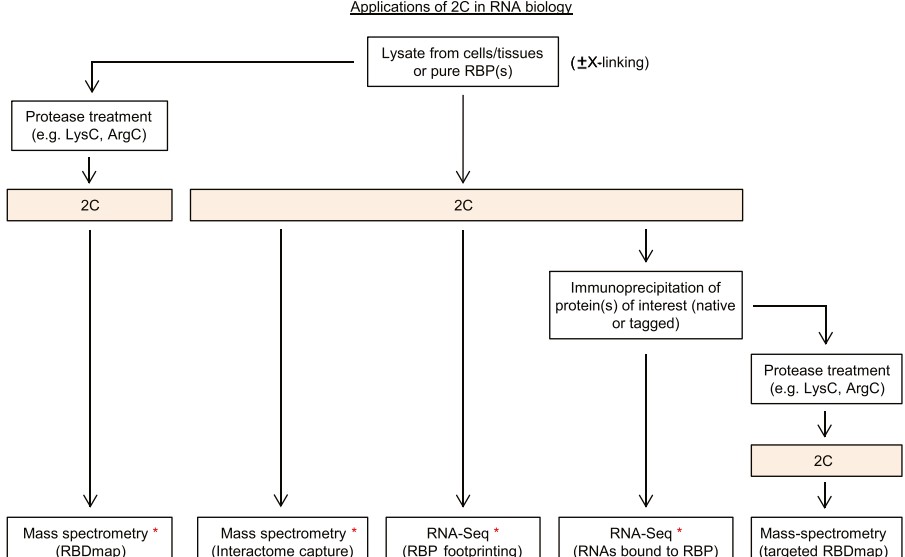

Applications of 2C in RNA biology

Figure 4.   Proposed applications of 2C in RNA biology.
For details, see the Discussion section. Also note that we envisage corresponding applications for DNA-binding proteins.

## Applications of 2C in RNA biology

We envisage the use of 2C to reach far beyond the detection and validation of RBPs shown here and to simplify several downstream applications to study RNA–protein interactions (Fig 4), both for the characterization of the RNA and the protein components. We suggest that 2C eluates could be directly analyzed by quantitative mass spectrometry in comparison with non–UV-treated background controls for the analysis of RNA-bound proteomes, effectively yielding simplified RNA interactome capture data not restricted to poly-adenylated RNAs. Furthermore, when samples are digested with proteases employed in RBDmap (Castello et al, 2016), such as ArgC or LysC, before 2C purification and eluates digested with trypsin before mass spectrometry, 2C could help to simplify the published RBDmap method (Castello et al, 2017). Moreover, a variation of this approach using immunoprecipitated individual RBPs or recombinant RBPs cross-linked to RNA could be used for targeted RBDmap.

From an RNA perspective, cross-link and immunoprecipitation protocols are frequently challenging for non-canonical RBPs. Because 2C enriches cross-linked proteins away from the "free" proteins, immunoprecipitation and RNA sequence analysis from 2C eluates could putatively benefit from sequencing libraries of higher quality. However, because 2C involves protein denaturation before RNP binding and elution, insufficient renaturation can interfere with immunoprecipitation by antibodies that recognize folded epitopes.

Moreover, 2C eluates could be used directly in protein–RNA footprinting experiments. Treatment of 2C eluates with RNase will preferentially degrade RNA that is not protected by cross-linked RBPs. After RNA digestion, the RBPs are degraded by proteinase K treatment, and the short RNA fragments that were protected from RNase treatment can be sequenced, generating a general map of RNA sequences bound by RBPs across the entire transcriptome.

Overall, the simplicity of the 2C method should also facilitate comparative analyses of multiple samples or conditions for the vast majority of the downstream applications discussed above.

Although this report focuses on RBPs and UV cross-linking as a means to induce covalent bonds between RNAs and RBPs, silica matrices can efficiently bind both DNA and RNA. Optimization of cross-linking procedures, lysis conditions, and buffers used for binding of the cross-linked nucleic acids to the silica matrices should further expand the utility to DNA-binding proteins and respective applications.

# Materials and Methods

### *S. cerevisiae* strains and manipulations

Standard methods were used for yeast culture, transformation, and manipulation (Amberg et al, 2005). WT BY4741 (*MATa*, *his3Δ1*, *leu2Δ0*, *met15Δ0*, and *ura3Δ0*) and PAB1-TAP (*MATa*, *leu2Δ0*, *met15Δ0*, *ura3Δ0*, and *PAB1-TAP::His3Mx6*) strains were purchased from Dharmacon. GAPDH-ProtA (*MATa*, *leu2Δ0*, *met15Δ0*, *ura3Δ0*, and *TDH3-ProtA::His3Mx6*) strain was generated in this study. A fragment of the TAP tag comprising the tobacco etch virus (TEV) cleavage target sequence, the two protein A tandem sequences, the *ADH1* terminator, and the *His3MX6* selection marker were amplified by PCR. Sequences complementary to the end of the *TDH3* gene were included by a second PCR, and the corresponding cassette was used to transform a WT strain for genomic integration of the TEV-Protein A cassette in frame with the *TDH3* gene.

### Yeast culture, cross-linking, and lysate preparation

Yeast cells were grown on YPD medium to mid-log phase (O.D.$_{600\ nm}$≈ 0.8–1.0), collected by centrifugation, resuspended in ice-cold buffer A (25 mM Tris–HCl, pH 7.5; 140 mM NaCl; 1.8 mM MgCl$_2$; and 0.01% NP-40), and transferred to 15-cm petri dishes on ice. Living cells were irradiated with 3 J/cm$^2$ of UV light at 254 nm in

a Spectrolinker XL-1500 (Spectronics Corporation) cross-linker. The cells were collected, concentrated by centrifugation, and the pellets were frozen until further use. Non-irradiated cells were processed in parallel as negative controls.

For cell lysis, pellets were thawed on ice and resuspended in ice-cold buffer B (25 mM Tris–HCl, pH 7.5; 140 mM NaCl; 1.8 mM MgCl$_2$; 0.5 mM DTT; and 0.01% NP-40) supplemented with protease inhibitors (Complete EDTA free #11873580001; Roche). The cells were broken by vortexing with 0.5-mm-diameter acid-washed glass beads (#G8772; Sigma-Aldrich) five times for 80 s, alternating with 1-min incubations on ice to avoid overheating of the samples. Lysates were clarified by centrifugation at 20,000 $g$ for 20 min and the supernatants were transferred to new tubes. Protein concentration was quantified by colorimetric assay (#5000006; Bio-Rad), and the samples were stored at –80°C until further use.

### Huh7 cell culture, cross-linking, and lysate preparation

Huh7 cells at 80%–90% confluency were washed twice with 10 ml chilled PBS on ice after media removal. Next, 20 ml chilled PBS was added on top of the cells, creating a thin liquid layer. The plates were placed on ice and irradiated with 150 mJ/cm$^2$ of UV light at 254 nm. Non-irradiated cells were processed in parallel as negative controls. The cells were harvested by scraping into PBS, pelleted, and resuspended in HMGN150 buffer (20 mM Hepes, pH 7.5; 150 mM NaCl; 2 mM MgCl$_2$; 0.5% NP-40; and 10% glycerol) supplemented with protease and RNase inhibitors. The cells were lysed on ice using a tip sonicator (Branson Sonifier Cell Disruptor B15) with the following settings: two cycles of 10 shots each at 50% duty cycle and output 4, with 10-s gap between the cycles. Cell debris and intact cells were spun down at 10,000 rpm, 4°C, for 10 min, and the clarified cell lysate was flash-frozen until further use.

### 2C method

Commercial buffers designed for RNA extraction were used for the 2C method. Although the exact composition of these buffers is undisclosed, they are based on the strong denaturing properties of chaotropic salts, which are widely used for RNA extraction (Gordon, 1972; Chirgwin et al, 1979; Chomczynski & Sacchi, 1987; Avison, 2008). In particular, lysis (6M guanidine thiocyanate, 4% sarcosyl, and 4% Titron X-100; #1060-1; Zymo Research), RNA prewash (4M guanidine hydrochloride and 80% ethanol; #1060-2; Zymo Research), and RNA wash (1% Tris–EDTA, pH 8, and 80% ethanol; #1060-3; Zymo Research) buffers were used. The whole protocol is conducted at room temperature and typically, the equivalent of 1 to 2 mg of protein from cell lysates were used per 2C extraction. One volume of lysate was combined with four volumes of lysis buffer. An equal volume of ethanol was added, the sample was mixed and applied to a Zymo-Spin V-E (#C1024; Zymo Research) silica column by vacuum, and the flowthrough was discarded. The column was transferred to a collection tube, and all subsequent steps were performed by centrifugation at 16,000 $g$ in a table-top microcentrifuge. The samples were consecutively treated with 400 $\mu$l of RNA prewash buffer and twice with RNA wash buffer. The column was then placed on top of a new microcentrifuge collection tube, and purified RNA and RNPs were eluted with 300 $\mu$l of water by centrifugation. The RNA

concentration was measured using NanoDrop 1000 (Thermo Fisher Scientific) and RNA integrity was assessed with Bioanalyzer RNA 6000 Nano Chips (#5067-1511; Agilent) (Schroeder et al, 2006).

### Silver staining of gels and Western blots

Between 20 and 30 $\mu$g of RNA from yeast or Huh7 cells were treated with or without 500 U of RNase I (#AM2295; Ambion) for 30 min at 30°C in a final volume of 50 $\mu$l. Laemmli loading buffer 4× (#1610747; Bio-Rad) supplemented with 2-mercaptoethanol (#M6250; Sigma-Aldrich) was added to each sample and boiled at 95°C for 5 min, and 25 $\mu$l were loaded onto 4%–15% Criterion SDS–PAGE gels (#5671084; Bio-Rad). Silver staining was performed according to the method of Mortz et al (2001). For Western blots, proteins from the complete gel, including the stacking gel and the loading wells, were transferred onto nitrocellulose membranes (#1704159; Bio-Rad) in a Trans-Blot Turbo Transfer System (#1704150; Bio-Rad) for 10 min. The following primary antibodies and dilutions were used to detect yeast proteins: anti-Pab1 1:4,000 (#ab189635; Abcam), anti-GAPDH 1:4,000 (#G9545; Sigma-Aldrich) (Silva et al, 2015), anti-histone H3 HRP 1:1,000 (#ab21054; Abcam) (Battaglia et al, 2017; de Vasconcellos et al, 2017), anti-hexokinase 1:10,000 (#4959-9988; Bio-Rad), anti-Tpi 1:4,000 (#10713-1-AP; Proteintech) (Winters et al, 2017), and anti-tubulin 1:4,000 (#ab6160; Abcam) (Young et al, 2013). The following antibodies were used to detect mammalian proteins: anti-hnRNPC1/C2 1:1,000 (#10294; GeneTex), anti-GAPDH 1:1,000 (#G9545; Sigma-Aldrich), anti-histone H3 HRP 1:10,000 (#ab21054; Abcam), and anti-FASTKD4 1:1,000 (#16245-1-AP; Proteintech). Anti-rabbit 1:10,000 (#ab95051; Abcam), anti-rat (#ab97057; Abcam), and anti-mouse 1:10,000 (#ab6789; Abcam and #NA931V; GE) HRP-conjugated secondary antibodies were used.

### Affinity purification

PAB1-TAP and GAPDH-ProtA strains were cross-linked as described and RBPs were purified by 2C. Non-irradiated samples from each strain were used as negative controls. Protein G–coupled Dynabeads (#10004D; Thermo Fisher Scientific) were incubated with rabbit IgG (#I5006; Sigma-Aldrich) for 30 min at room temperature. Unbound IgG was eliminated by washing the beads three times with buffer B, and fractions from the 2C eluates equivalent to 100 $\mu$g of RNA were incubated with the beads with gentle rotation at 4°C for 2 h. Unbound material was removed, and the beads were washed once with buffer B, three times with buffer C (25 mM Tris–HCl, pH 7.5; 1 M NaCl; 1.8 mM MgCl$_2$; 0.5 mM DTT; and 0.01% NP-40), and another time with buffer B. Pab1-TAP was eluted by incubating the beads in buffer D (25 mM Tris–HCl, pH 7.5; 140 mM NaCl; 1.8 mM MgCl$_2$; 0.5 mM DTT; and 0.01% NP-40) with 2.5 $\mu$g of TEV protease at 34°C for 1 h and GAPDH-ProtA was eluted by resuspending the beads in buffer D plus loading buffer and incubating them at 95°C for 5 min.

## Acknowledgements

We thank Ina Huppertz and Mai Sun for experimental support, and all members of the Hentze lab for their helpful discussion and advice. This work

was supported by the European Research Council Advanced Grant ERC-2011-ADG_20110310 to MW Hentze.

## Author Contributions

C Asencio: conceptualization, resources, formal analysis, validation, investigation, methodology, and writing—original draft, reviewing, and editing.
A Chatterjee: formal analysis, validation, and investigation.
MW Hentze: conceptualization, resources, formal analysis, supervision, funding acquisition, and writing—original draft, project administration, and writing—reviewing and editing.

## Conflict of Interest Statement

The authors declare that they have no conflict of interest.

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
