## [Reviewer comments · Life Science Alliance]

Silica-based solid phase extraction of crosslinked nucleic acid-bound proteins

Claudio Asencio, Aindrila Chatterjee and Matthias W. Hentze

DOI: 10.26508/lsa.201800088

Review timeline:	Submission date:	11 May 2018
	1 st Editorial Decision:	1 June 2018
	1 st Revision received:	5 June 2018
	2 nd Editorial Decision:	6 June 2018
	2 nd Revision Received:	6 June 2018
	Accepted:	6 June 2018

Report:

(Note: Letters and reports are not edited. The original formatting of letters and referee reports may not be reflected in this compilation.)

1st Editorial Decision

1 June 2018

Thank you for submitting your manuscript entitled "Silica-based solid phase extraction of nucleic acid-bound proteins" to Life Science Alliance. The manuscript was assessed by expert reviewers, whose comments are appended to this letter.

As you will see, all reviewers think that your method is very valuable to the field. They provide constructive input on how to further strengthen your manuscript and point out that a few clarifications are needed, many of which can be addressed in the text. I would thus like to invite you to provide a revised version of your manuscript, addressing the individual points made by the reviewers in a minor revision.

Thank you for this interesting contribution to Life Science Alliance. We are looking forward to receiving your revised manuscript.

REPORTS

Reviewer #1 (Comments to the Authors (Required)):

The authors report that a simple, established method for RNA purification is also suitable for purifying proteins covalently cross-linked to RNA and their subsequent analysis. The method is convincingly validated and should be useful. I recommend publication of the paper.

Main comments:

1. Description of the method should be more detailed (lines 245ff). Was the procedure performed at room temperature? Report the volumes of buffer used for the various steps. The initial sample application was done by vacuum. Is the same true for washing and elution steps? The elution step mentions a 'microcentrifuge collection tube', so perhaps washing steps were performed by centrifugation rather than vacuum? Was there

any incubation time between application of the buffer and its removal by vacuum or centrifugation? If centrifugation was performed, report the conditions. What does 'elution into water' mean (lines 83 and 256)? I assume that water was used as an eluent, but the wording suggests that the collection tube contained water.

2. Fig. 1C shows some background in the non-crosslinked sample. Please discuss if this level would be a problem in MS analysis of the total column-bound material and if the background can be reduced by additional washes.

3. Lines 112ff: The authors argue that the comparison between input and silica column eluate allows some conclusions regarding the 'potency' of RNA binding. This is not the case. The authors do not take cross-linking efficiency into account, which varies widely between RNA binding proteins. A weak binder with good cross-linking efficiency may look more 'potent' than a good binder that cross-links poorly.

4. Lines 116/117: 'specificity of detection of both PABPN1 and GAPDH as RBPs': I am not convinced. Under the circumstances discussed, 'specificity' would be limited to a comparison of DNA and RNA to begin with. Even this comparison cannot really be made, as there is a large excess of RNA over DNA in the cell, and only a minute fraction of the DNA will be single-stranded. (Regarding GAPDH, the ligand that RNA should be compared with under biological circumstances would be NAD!)

5. Line 133: 'Immunoprecipitation experiments' is misleading, as generic IgG was used to precipitate protein A-tagged proteins. This is not what one usually calls an immunoprecipitation. Whether a 'real' IP works after the silica column will presumably depend on the particular antigen/antibody combination.

Minor points:

Title: 'nucleic acid-bound proteins' is misleading. The method will only work for COVALENTLY bound proteins.

Abstract, line 21: The word 'discovery' is perhaps too grand. Since the silica matrix binds RNA, it would have been safe to predict that it will also retain any molecule covalently bound to the RNA. (There are other reasons why the method might not have worked, so it will be useful to have this paper.)

Line 59: References for 'interactome capture' are a bit Hentze-centric; other labs have performed similar work.

In my opinion, the method is so straightforward that Fig. 1A could be omitted.

Line 103: I doubt that the word 'specifically' is justified here.

Lines 175ff: Delete the sentence starting '2C should also increase....'

I find the discussion under 'Applications of 2C in RNA biology' quite sufficient and suggest deleting Fig. 4.

I wonder if the method described is so unexpected, complex.....that it requires an extra name ('2C').

Reviewer #2 (Comments to the Authors (Required)):

Silica-based solid phase extraction of nucleic acid-bound proteins

In this manuscript, Asencio et al. report on a short and efficient method allowing for the analysis of UV cross-linked RNPs in a single step. It is known that silica-based matrices efficiently bind nucleic acids due to the negative charge of their phosphate backbones and the positively charged silica particles. This principle is used in many DNA/RNA extraction kits. Here, the authors show that not only nucleic acids (here RNAs) can be isolated with such an approach but also cross-linked proteins or protein fragments. The authors analyze the eluted RNA/RNPs by silver staining, western blotting of known RNA binding proteins and also perform immunoprecipitation of the eluted sample for further downstream analysis. They refer to their method as 'complex capture' or '2C'.

This is a very short rather technical manuscript demonstrating that known silica-base nucleic acid purification methods can also be used for RNP enrichment. Since RNA-protein interactions are widely studied, improvements in isolation methods making it more accessible for a broader part of the scientific community are highly valuable. Although the study does not go very far, it has its merits. It is well written and presented. I have only a few comments that are listed below.

1. The interaction between RNA and the silica matrix is based on the charge of the molecules. Heavily phosphorylated (hyper-phosphorylated) proteins might also accumulate locally high negative charge and thus might bind to the matrix as well. Has this been tested? The authors should test that.
2. How would the input RNA in Figure 1B look like? This could be added.
3. Is the TAP-tag cleaved off after the elution in Figure 3A? The lower band seems to be enriched or is there any other explanation for the different migration?

Reviewer #3 (Comments to the Authors (Required)):

The authors present an interesting observation that silica columns can be used to purify crosslinked RNA-protein complexes.

The following key claims are made in the paper and, for the most part, are well supported by the experimental data provided. Specifically, that the Complex Capture (2C) method provides a silica based enrichment for crosslinked nucleic acid-bound proteins. As a proof of concept isolated cells from yeast and mammalian cells and demonstrated enrichment of known RNA binding proteins and depletion of control proteins. The enrichment step makes RNA labeling redundant. Finally, they showed that the purified RNPs can be immunoprecipitated.

I believe that this method is a very important contribution for RNP analysis and will provide an important technical advance that I expect will be widely used. I think it should be published in Life Science Alliance.

There are a few minor points that I think the authors should address or clarify prior to publication.

Minor comments:

Why do the authors think that in Figure 1C non-crosslinked and crosslinked sample show the same banding pattern? This suggests that the enrichment is based off of affinity to the SILICA matrix and not necessary just from crosslinked RNA.

In Figure 1A, the authors state that "These nucleic acid-binding proteins could later be co-eluted with pure DNA, RNA or both (Figure 1A)." They mention DNA but no data with DNA presented. One of the main points of their paper is that the enrichment makes RNA labeling redundant. There are several counterexamples where this can be refuted. For example in Figure 1 the fact that they see several proteins in the -RNase lanes which do not shift in position with +RNase treatment. This suggests that these proteins are NOT crosslinked to RNA and are instead migrating at their known molecular weight. This should be clarified or explained.

In Figure 2, what are all the extra bands in hexokinase 2C no crosslinked and crosslinked lanes? The input and 2C selected samples (in both non-crosslinked and crosslinked lanes) have a vastly different molecular weight that is resistant to RNase treatment suggesting it is NOT crosslinked to RNA.

GAPDH, TPi and hexokinase signal not enriched at input molecular weight by RNase treatment in yeast cells unlike hnrnpC and PABP1. This suggests that these are not actually crosslinked to RNA or that RNA labeling is in fact NOT redundant to the 2C selection. These points should be clarified.

Figure 3 presents a nice proof of concept example of downstream applicability with purified complexes but little information is given regarding the differences between this and non-2C selected samples. Because the elution fraction is not stated the yield looks pretty low from the 2C immunoprecipitations, that is there is not more signal in the elution than in the input which is what you would expect for any enrichment step. Is the 2C affecting the immunoprecipitation yield?

1st Revision – authors' response

5 June 2018

Reviewer #1 (Comments to the Authors (Required)):

The authors report that a simple, established method for RNA purification is also suitable for purifying proteins covalently cross-linked to RNA and their subsequent analysis. The method is convincingly validated and should be useful. I recommend publication of the paper.

Main comments:

1. Description of the method should be more detailed (lines 245ff). Was the procedure performed at room temperature? Report the volumes of buffer used for the various steps. The initial sample application was done by vacuum. Is the same true for washing and elution steps? The elution step mentions a 'microcentrifuge collection tube', so perhaps washing steps were performed by centrifugation rather than vacuum? Was there any incubation time between application of the buffer and its removal by vacuum or

centrifugation? If centrifugation was performed, report the conditions. What does 'elution into water' mean (lines 83 and 256)? I assume that water was used as an eluent, but the wording suggests that the collection tube contained water.

The description of the method has been improved by the addition of further technical details, as suggested by the reviewer.

2. Fig. 1C shows some background in the non-crosslinked sample. Please discuss if this level would be a problem in MS analysis of the total column-bound material and if the background can be reduced by additional washes.

Since mass spec-based analyses quantitatively compare signals in the crosslinked versus the non-crosslinked samples, the modest level of background seen in the non-crosslinked samples would in our experience not pose a significant problem. Nonetheless, if required, it is possible to further reduce the background in different ways. Pre-incubation of the lysates at 70°C for 5 minutes, which dissociates non-crosslinked RNA-protein complexes, results in basically no visible protein contamination in a silver gel in the non-crosslinked sample. We have now included this information in the revised manuscript. A second alternative is to perform two consecutive rounds of 2C. This usually reduces the overall yield, but the level of protein content in the non-crosslinked sample is virtually undetectable in a silver gel. Thus, depending on the intended downstream application for the enriched RNA-protein complexes, the conditions of 2C can be adjusted for either higher yields or further reduction of background noise in the non-crosslinked samples.

3. Lines 112ff: The authors argue that the comparison between input and silica column eluate allows some conclusions regarding the 'potency' of RNA binding. This is not the case. The authors do not take cross-linking efficiency into account, which varies widely between RNA binding proteins. A weak binder with good cross-linking efficiency may look more 'potent' than a good binder that cross-links poorly.

We agree with this comment and we have amended the discussion of this point in the revised manuscript.

4. Lines 116/117: 'specificity of detection of both PABPN1 and GAPDH as RBPs': I am not convinced. Under the circumstances discussed, 'specificity' would be limited to a comparison of DNA and RNA to begin with. Even this comparison cannot really be made, as there is a large excess of RNA over DNA in the cell, and only a minute fraction of the DNA will be single-stranded. (Regarding GAPDH, the ligand that RNA should be compared with under biological circumstances would be NAD!)

We have reworded this paragraph in the revised manuscript.

5. Line 133: 'Immunoprecipitation experiments' is misleading, as generic IgG was used to precipitate protein A-tagged proteins. This is not what one usually calls an immunoprecipitation. Whether a 'real' IP works after the silica column will presumably depend on the particular antigen/antibody combination.

To avoid any misinterpretation, we have rephrased this paragraph, referring to affinity purification now. We agree that immunoprecipitations sensu strictu would depend on the particular antigen/antibody combinations.

Minor points:

Title: 'nucleic acid-bound proteins' is misleading. The method will only work for COVALENTLY bound proteins.

The title has been modified accordingly.

Abstract, line 21: The word 'discovery' is perhaps too grand. Since the silica matrix binds RNA, it would have been safe to predict that it will also retain any molecule covalently bound the RNA. (There are other reasons why the method might not have worked, so it will be useful to have this paper.)

Abstract has been edited and the word 'discovery' has been substituted by 'observation'. Nonetheless, we like to state that the "observation" at hand has been missed for decades, which would justify the use of the term "discovery" in our opinion.

Line 59: References for 'interactome capture' are a bit Hentze-centric; other labs have performed similar work.

As far as we are aware, many of the published reports come from our lab, either directly or as collaborations, resulting in the unintended effect noticed by the reviewer. Nonetheless, we appreciate the comment, as it alerted us to the missing inclusion of one of the original reports by Baltz et al., which has now been added.

In my opinion, the method is so straightforward that Fig. 1A could be omitted.

The method is indeed conceptually simple, and for this methods-oriented paper, a graphical representation of the method could be helpful for some of the readership. Consequently, we have retained Fig. 1A, but we would be prepared to remove it, if editorially requested.

Line 103: I doubt that the word 'specifically' is justified here.

The word 'specifically' has been removed from this sentence.

Lines 175ff: Delete the sentence starting '2C should also increase....'

This sentence has been deleted.

I find the discussion under 'Applications of 2C in RNA biology' quite sufficient and suggest deleting Fig. 4.

Figure 4 graphically summarizes and logically organizes the variety of downstream applications of 2C. We suggest retaining it in the manuscript for the readers' benefit. However, we would be prepared to remove it, if editorially preferred.

I wonder if the method described is so unexpected, complex.....that it requires an extra name ('2C').

It has become customary to refer to procedures and methods by a short name, often an acronym. Our proposed name is intended to reflect the simplicity of the method and provides an easy way to differentiate it from other, more laborious alternatives to purify nucleic acid-protein complexes. While we would not refuse removal of the name, if editorially preferred, we suggest to maintain it for the sake of simple future reference.

Reviewer #2 (Comments to the Authors (Required)):

Silica-based solid phase extraction of nucleic acid-bound proteins

In this manuscript, Asencio et al. report on a short and efficient method allowing for the analysis of UV cross-linked RNPs in a single step. It is known that silica-based matrices efficiently bind nucleic acids due to the negative charge of their phosphate backbones and the positively charged silica particles. This principle is used in many DNA/RNA extraction kits. Here, the authors show that not only nucleic acids (here RNAs) can be isolated with such an approach but also cross-linked proteins or protein fragments. The authors analyze the eluted RNA/RNPs by silver staining, western blotting of known RNA binding proteins and also perform immunoprecipitation of the eluted sample for further downstream analysis. They refer to their method as 'complex capture' or '2C'.

This is a very short rather technical manuscript demonstrating that known silica-base nucleic acid purification methods can also be used for RNP enrichment. Since RNA-protein interactions are widely studied, improvements in isolation methods making it more accessible for a broader part of the scientific community are highly valuable. Although the study does not go very far, it has its merits. It is well written and presented. I have only a few comments that are listed below.

1. The interaction between RNA and the silica matrix is based on the charge of the molecules. Heavily phosphorylated (hyper-phosphorylated) proteins might also accumulate locally high negative charge and thus might bind to the matrix as well. Has this been tested? The authors should test that.

This has not been directly tested so far. Since retention of RNA is based on its negative charge, we concur with the reviewer that proteins with a very high net negative charge density could also be retained without being crosslinked to RNA. These proteins are expected to be quite rare, and their electrophoretic migration would be unaffected by RNase treatment, unlike the RBPs shown here. Moreover, since 2C enriches for RBPs and helps to define candidate RBPs, any specific protein requires independent validation anyhow.

2. How would the input RNA in Figure 1B look like? This could be added.

Input RNA is identical to the 'NoCL' sample shown in blue. It is, therefore, already included.

3. Is the TAP-tag cleaved off after the elution in Figure 3A? The lower band seems to be enriched or is there any other explanation for the different migration?

Pab1-TAP was eluted by TEV cleavage and as a consequence, the majority of the eluted protein migrates faster to a lower position in the gel. However, the incubation of the beads at 34°C for 1h during the TEV protease digestion can also result in some minor release of undigested Pab1-TAP from the beads, resulting in the low intensity band at the same molecular weight as the input protein.

GAPDH-ProtA was eluted by boiling the beads and nothing has been cleaved off proteolytically. Therefore, the migration is unchanged. The elution conditions for each protein are now better explained in the materials and methods section.

Reviewer #3 (Comments to the Authors (Required)):

The authors present an interesting observation that silica columns can be used to purify crosslinked RNA-protein complexes.

The following key claims are made in the paper and, for the most part, are well supported by the experimental data provided. Specifically, that the Complex Capture (2C) method provides a silica based enrichment for crosslinked nucleic acid-bound proteins. As a proof of concept isolated cells from yeast and mammalian cells and demonstrated enrichment of known RNA binding proteins and depletion of control proteins. The enrichment step makes RNA labeling redundant. Finally, they showed that the purified RNPs can be immunoprecipitated.

I believe that this method is a very important contribution for RNP analysis and will provide an important technical advance that I expect will be widely used. I think it should be published in Life Science Alliance.

There are a few minor points that I think the authors should address or clarify prior to publication.

Minor comments:

Why do the authors think that in Figure 1C non-crosslinked and crosslinked sample show the same banding pattern? This suggests that the enrichment is based off of affinity to the SILICA matrix and not necessary just from crosslinked RNA.

In our analysis, the pattern of the 'input' samples and the 'NoCL' eluates resemble each other. This resemblance suggests that a small amount of the abundant cellular proteins contaminates the eluates, as the most prominent bands can also be seen in the 'CL' eluates. However, the 'CL' eluates show a predominantly different overall pattern with numerous bands that are absent from the other lanes. This is also evidenced by the Western blot analyses. As commented above in response to main comment 2 of reviewer #1, contamination can be further reduced, if analytically required.

In Figure 1A, the authors state that "These nucleic acid-binding proteins could later be co-eluted with pure DNA, RNA or both (Figure 1A)." They mention DNA but no data with DNA

presented. One of the main points of their paper is that the enrichment makes RNA labeling redundant. There are several counterexamples where this can be refuted. For example in Figure 1 the fact that they see several proteins in the -RNase lanes which do not shift in position with +RNase treatment. This suggests that these proteins are NOT crosslinked to RNA and are instead migrating at their known molecular weight. This should be clarified or explained.

This manuscript focusses on applications of 2C in RNA biology. However, it is known that silica matrices can bind DNA and RNA, depending on the buffer conditions. We discuss potential applications of 2C for the isolation of DNA-binding proteins.

The key criterion for the validation of RBPs is their absence from non-crosslinked samples. The differential migration upon treatment with RNase and/or an increase in the intensity of the main band in RNase-treated samples can be used as additional criteria. However, in cases where the molecular mass of the RBP is large in comparison to the crosslinked RNA, potentially also due to (chemical) degradation of the RNA in samples that were not treated with exogenous RNase, no such shifts can be observed.

In Figure 2, what are all the extra bands in hexokinase 2C no crosslinked and crosslinked lanes? The input and 2C selected samples (in both non-crosslinked and crosslinked lanes) have a vastly different molecular weight that is resistant to RNase treatment suggesting it is NOT crosslinked to RNA.

Saccharomyces cerevisiae hexokinase B is known to aggregate under denaturing conditions in vitro into amyloid-like fibrils (Ramshini et al., PLoS One, January 13, 2011). The denaturing conditions during 2C capture thus may have promoted the formation of hexokinase aggregates resistant to SDS-PAGE separation, although it cannot be excluded that the extra bands are a consequence of cross-reactivity of the antibody with another yeast (RNA-binding) protein.

GAPDH, TPI and hexokinase signal not enriched at input molecular weight by RNase treatment in yeast cells unlike hnrnpC and PABP1. This suggests that these are not actually crosslinked to RNA or that RNA labeling is in fact NOT redundant to the 2C selection. These points should be clarified.

Please see our comment and explanation above.

Figure 3 presents a nice proof of concept example of downstream applicability with purified complexes but little information is given regarding the differences between this and non-2C selected samples. Because the elution fraction is not stated the yield looks pretty low from the 2C immunoprecipitations, that is there is not more signal in the elution than in the input which is what you would expect for any enrichment step. Is the 2C affecting the immunoprecipitation yield?

Figure 3 shows that the affinity purifications of Pab1 and GAPDH are specific for the respective tagged proteins. About 8% of the eluates after the IPs were loaded for analysis. This indicates that there is strong enrichment in the eluate fractions after the affinity purification. However, we fully concur and explicitly discuss in the manuscript that 2C can

affect the efficiency of downstream methods. As discussed in the text and in response to point 5 from reviewer #1, it should be taken into consideration that the 2C method involves protein denaturation and therefore, the success of any downstream procedure will depend on the requirements of any particular antigen-antibody combination.

2nd Editorial Decision

6 June 2018

Thank you for submitting your revised manuscript entitled "Silica-based solid phase extraction of crosslinked nucleic acid-bound proteins". I appreciate the changes introduced during revision. I am thus happy to publish your paper in Life Science Alliance pending final revisions necessary to meet our formatting guidelines. Congratulations!